# Efficient Learning rate schedules for Stochastic Non-negative Matrix Factorization via Reinforcement Learning

**Shreyas Subramanian, Vignesh Ganapathiram, & Aly El Gamal** [*]
Amazon.com, Inc., 410 Terry Ave N, Seattle 98109, WA
{subshrey,vignesga,alyeg}@amazon.com

## Abstract

For deep learning training, learning rate schedules are often picked through trial and error, or hand-crafted optimization algorithms that focus mostly on maintaining stability and convergence without systemic incorporation of higher order derivative information to optimize the convergence slope. In this paper, we consider a stochastic version of Non-negative Matrix Factorization (NMF) where only a noisy gradient is known, and calculate a theoretical upper bound for SGD learning rate (LR) schedule that guarantees convergence, thereby providing a clean example where stability and convergence is not a challenge. We then use a Reinforcement Learning agent to demonstrate how efficient LR schedules, superior to those found by traditional algorithms, can be found for this NMF problem.

## 1 Introduction

Current algorithms for optimizing the training of deep neural networks typically rely on using shallow gradient information (only first order in most cases) to consistently seek stable trajectories towards good local optima. There are only few exceptions that attempt to infer useful higher order gradient information to optimize the convergence rate (see (Goodfellow et al., 2016, Chapter 8)) and even those use a fixed strategy (policy) to do so. In this work, we consider a simple setup where only the learning rate (LR) schedule is being optimized and stability is not difficult to attain. In particular, we consider the Non-negative Matrix Factorization (NMF) problem. For a matrix $V \in \mathbb{R}_+^{m \times n}$, NMF finds two matrices $W \in \mathbb{R}_+^{m \times r}$ and $H \in \mathbb{R}_+^{r \times n}$ such that $V \approx WH$. NMF has been applied to many applications like recommendation systems Luo et al. (2014); Khan et al. (2020); Li et al. (2009a;b); Luo et al. (2015), image decomposition Tang et al. (2013); Zhao et al. (2008); Kalayeh et al. (2014); Padilla et al. (2011); Zhang et al. (2004), and acoustic signals Kameoka et al. (2009); Févotte et al. (2018); Hennequin et al. (2010); Yoshii et al. (2013).

We consider SGD optimization for a stochastic version of NMF where only noisy information about the cost gradient is accessible. In particular, the matrix decomposition is found by using gradient descent to minimize the distance measure $\|V - WH\|_F^2$ where $\|\cdot\|_F$ is the Frobenius norm. Each factor is alternatively updated using the gradient and a learning rate $\alpha$ (or schedule). For example, for $W$ updates and loss function $F$, we can find $W^{t+1} \approx (W^t - \alpha \nabla_W F(W^t, H^t))$. Using the Frobenius norm above, a commonly used rule to select step length (see Guan et al. (2011) for example) is the Armijo rule, where the step size $\alpha$ is iteratively calculated as a result of the Armijo line search Zhang et al. (2006). At each iteration $t$ we aim to find the maximum stable learning rate, e.g., maximum $\alpha^t$ that satisfies (for the $W$ update) $F(W^{t+1}, H) \leq F(W^t, H) - c \cdot \alpha^t \|\nabla F(W^t, H)\|^2$. In this work, we first show that guaranteeing stability is straightforward as long as $\alpha$ is below a certain threshold. Confined to the stable region, we demonstrate empirically how traditional algorithms perform slightly better, on average, than randomly selecting $\alpha$, and that superior performance is consistently delivered by an RL agent that adjusts its policy in a data-driven fashion to determine $\alpha^t$ in every iteration $t$. We conjecture that the agent implicitly infers useful second-order derivative information via this policy adjustment process.

---
[*]

## 2 THEORETICAL BOUNDS ON CONVERGENT LR SCHEDULES FOR STOCHASTIC NMF

We assume iterative SGD with unbiased noisy gradients, and identify theoretical bounds for stable LR schedules. Theorem A.3 states that the optimal LR schedule $(\alpha^t)$ for NMF with SGD is bounded between $0 < \alpha^t < \frac{2}{L}$, where $L = \|W^t\|^2$ when updating $W$ with a fixed $H$ (with symmetry for the other case). Any LR schedule within this bound will lead to convergence as we demonstrate empirically below. Detailed assumptions and proofs are included in Appendix A and Appendix B.

## 3 RESULTS: STOCHASTIC NMF SGD WITH RL SCHEDULE

We ran experiments for NMF with a randomly and sparsely initialized matrix of size $10000 \times 1000$ with a latent dimension of $100$ for SGD and several fixed LRs. Figure 1a shows loss curves for fixed LRs from $1e-5$ to $10$. Corresponding theoretical upper bound LRs are calculated based on Theorem A.3 as the problem converges (shown in Figure 1b). We conducted 25 trials where the LR for the current epoch is calculated as a random value less than the upper bound for that LR. Interestingly, we also observe that the loss curves from these random schedules closely follow the loss curves corresponding to the mean LR (see loss curves in Appendix B).

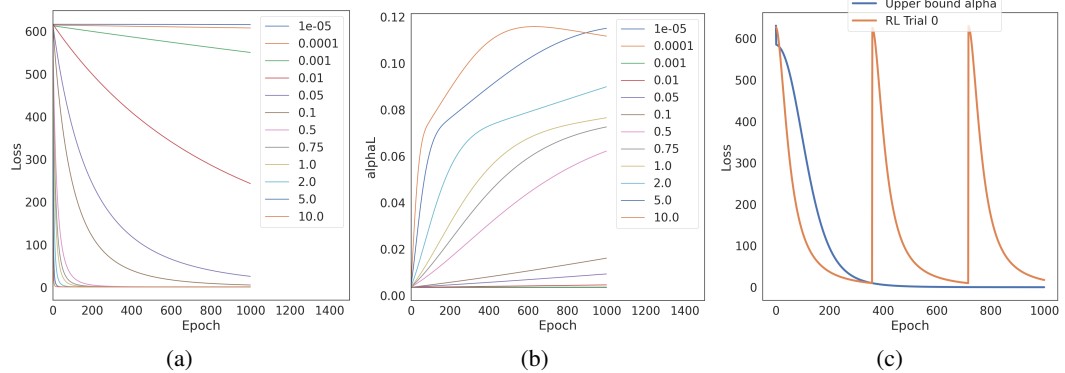

Figure 1: a) Fixed LR loss curves b) Corresponding Upper bound (Theorem A.3) c) Learned schedule vs. a fixed upper-bound loss.

We consider an RL agent whose reward reflects improvement over best encountered loss and number of remaining epochs (details in Appendix C). We trained the agent using Proximal Policy Optimization (PPO) Schulman et al. (2017) to find LR schedules within bounds of $[0, 1]$ for NMF with SGD with observations of the current loss value, epoch number, and gradient norm[1]. The agent is penalized with a high artificial loss on divergence and the episode is reset. As we can see in Fig. 1c, the RL agent finds a schedule with a very high convergence rate, and is seen resetting and converging thrice (can be handled in practice via early stopping mechanisms) within the 1000 epoch run, compared to the run with theoretical upper bound loss curve (more experiments and action histories are in Appendix C).

## 4 CONCLUSION

We identifed theoretical upper bound for LR schedules for stochastic NMF convergence with SGD, and ran experiments with fixed, random and learned RL1 agent schedules; showing that it is possible to learn efficient problem-specific schedules for typical gradient-based learning problems. Our follow-up empirical work with more complex deep learning problems also show promising results with the same method presented here.

---

[1]The upper bound of 1 used here is 10 times the theoretical upper bound shown in Fig 1b. However in practice we see the optimal agent suggesting low values of LR.

URM STATEMENT

The authors acknowledge that at least one key author of this work meets the URM criteria of ICLR 2023 Tiny Papers Track.

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

## A    APPENDIX A - THEORETICAL BOUNDS ON OPTIMAL LR SCHEDULES

First, we show that $F$ is convex with respect to (w.r.t.) $W$ or $H$ alone. Since $F$ is symmetric and the order of updates does not matter, we can show that

**Lemma A.1.** $F(W^t, H)$ *is convex w.r.t. $W$ when $H$ is fixed, and convex w.r.t. $H$ when $W$ is fixed.*

*Proof.* For three matrices $H_1$, $H_2$ and $H_\lambda \in \mathbb{R}_+^{r \times n}$, and a positive number $\lambda \in (0, 1)$, let:

$$H_\lambda = \lambda H_1 + (1 - \lambda)H_2,$$

then we have the following,

$$
\begin{aligned}
F(W^t, H_\lambda) &= F(W^t, \lambda H_1 + (1 - \lambda)H_2) \\
&= \frac{1}{2}\|V - W^t(\lambda H_1 + (1 - \lambda)H_2)\|_F^2 \\
&= \frac{1}{2}\operatorname{tr}\left(V - W^t(\lambda H_1 + (1 - \lambda)H_2)\right)^\top \left(V - W^t(\lambda H_1 + (1 - \lambda)H_2)\right) \\
&= \frac{1}{2}\sum_{ij}\left(V_{ij} - W_{ij}^t\left(\lambda H_{1,ij} + (1 - \lambda)H_{2,ij}\right)\right)^2 \\
&= \frac{1}{2}\sum_{ij}\left(V_{ij} - \lambda W_{ij}^t H_{1,ij} - (1 - \lambda)W_{ij}^t H_{2,ij}\right)^2 \\
&= \frac{1}{2}\sum_{ij}\left(V_{ij} + \lambda V_{ij} - \lambda W_{ij}^t H_{1,ij} - (1 - \lambda)W_{ij}^t H_{2,ij} - \lambda V_{ij}\right)^2 \text{ (Adding and subtracting } \lambda V_{ij}) \\
&= \frac{1}{2}\sum_{ij}\left(\lambda\left(V_{ij} - W^t H_{1,ij}\right) + (1 - \lambda)\left(V_{ij} - W^t H_{2,ij}\right)\right)^2 \\
&\leq \frac{\lambda}{2}\sum_{ij}\left(V_{ij} - W^t H_{1,ij}\right)^2 + \frac{(1 - \lambda)}{2}\sum_{ij}\left(V_{ij} - W^t H_{2,ij}\right)^2 \\
&\leq \lambda \cdot F(W^t, H_1) + (1 - \lambda) \cdot F(W^t, H_2).
\end{aligned}
$$

Therefore, $F$ is convex when updating $H^t$. Similarly we can prove that it is convex when updating $W^t$.

$\square$

**Lemma A.2.** *The gradient of $F(W^t, H)$ is Lipschitz continuous when updating $W^t$ while fixing $H$ with a Lipschitz constant of $L = \|(W^t)^\top W^t\|$, and is Lipschitz continuous when updating $H^t$ when $W$ is fixed with a Lipschitz constant of $\|L = (H^t)^\top H^t\|$.*

*Proof.* Because of symmetry, we only prove the first part of the statement. We know that the gradient of $F$

$$\nabla F(W^t, H) = (W^t)^\top(W^t H - V).$$

Therefore,

$$
\begin{aligned}
\|\nabla F(W^t, H) - \nabla F(W^{t+1}, H)\| &= \|(W^t)^\top W^t(W^t - W_{t+1})\| \\
&\leq L\|(W^t - W_{t+1})\|,
\end{aligned}
$$

where $L = \|(W^t)^\top W^t\|$.

$\square$

We now provide precise statements for the considered assumptions and then the main theorem statement and its proof.

**Assumption A.1.** *In the stochastic NMF setting, assume that we only have access to noisy gradient estimates and not the actual gradient, and that $E[\nabla F_i(W^t, H)] = \nabla F(W^t, H) \; \forall$ samples $i$.*

**Assumption A.2.** *Assume that the simplest form of Stochastic Gradient Descent (SGD) is used to find subsequent iterates of $W$ and $H$. Focusing on $W$, we have the following update formula:*

$$W^{t+1} = W^t - \alpha^t \cdot \nabla F(W^t, H)$$

*where $t$ is the iteration counter.*

Typically, step size selection is through heuristics or experimentation. The most common way to select step size is to use a constant throughout the training process, i.e.:

$$\alpha^t = \alpha_0.$$

For NMF problems, another typical way to select the step size is through Armijo line search. At each iteration $t$ we aim to find an $\alpha^t$ that satisfies:

$$F(W^{t+1}, H) \leq F(W^t, H) - c \cdot \alpha^t \|\nabla F(W^t, H)\|^2. \tag{1}$$

That is, at each iteration $t$, we start with a large $\alpha_{max}$ and successively try step sizes with diminishing multiplicative factor until we find the largest $\alpha$ that satisfies the above equation. Additionally, by definition, 1 satisfies the Polyak-Lojasiewicz (PL) inequality at convergence, where for $\mu > 0$ and function $F$:

$$\frac{1}{2}\|\nabla F(W^t, H)\|^2 \geq \mu(F(W^t, H) - F^*).$$

From 1, we can expect a sequence of $\mu^t$ where $\mu^t = \frac{1}{2c\alpha^t}$ . We can therefore expect that F decreases towards its optimal value $F^*$ with the appropriate selection of $\alpha$. This leads us to the following theorem:

**Theorem A.3.** *For SGD to converge for the NMF problem, we select $\alpha^t$ such that $0 < \alpha^t < \frac{2}{L}$ where $L = \|W^t\|^2$ when updating $W^t$ while fixing $H$, and $L = \|H^t\|^2$ when updating $H^t$ and fixing $W$. As a consequence, Armijo line search leads to convergence for any constant $0 < c < 1$.*

*Proof.* From Lemma 1.1 and 1.2, we know that $F$ is convex and $\nabla F$ is Lipschitz continuous w.r.t. each of the two parameters when the other is fixed.

Focusing just on the $W^t$ update step, and dropping $H$ notation in $F$ while using the Lipschitz condition and SGD update rule, we have:

$$F(W^{t+1}) \leq F(W^t) + \left(\nabla F(W^t)\right)^\top (W^{t+1} - W^t) + \frac{L}{2}\|W^{t+1} - W^t\|^2$$

$$\leq F(W^t) + \left(\nabla F(W^t)\right)^\top (-\alpha^t \nabla F(W^t)) + \frac{L}{2}\|-\alpha^t \nabla F(W^t)\|^2$$

$$\leq F(W^t) - \|F(W^t)\|^2 \left(\alpha^t - \alpha^{t^2}\frac{L}{2}\right)$$

$$\implies F(W^t) \geq F(W^{t+1}) + \|F(W^t)\|^2 \left(\alpha^t - \alpha^{t^2}\frac{L}{2}\right)$$

For $F$ to decrease towards $F^* = \min_{W,H} F(W, H)$, we need:

$$\alpha^t - \alpha^{t^2}\frac{L}{2} > 0$$

$$\implies \alpha^t(2 - \alpha^t L) > 0$$

$$\implies 0 < \alpha^t < \frac{2}{L},$$

where $L = \|(W^t)^\top W^t\|$. □

Lastly, when using Armijo line search in 1, we can see that :

$$c\alpha^t \leq \alpha^t - \alpha^{t^2}\frac{L}{2}$$
$$\implies c \leq 1 - \alpha^t\frac{L}{2}$$
$$\implies \alpha^t \leq \frac{2(1-c)}{L}.$$

With $0 < \alpha^t < \frac{2}{L}$, we have:

$$0 < c < 1. \tag{2}$$

## B   APPENDIX B - LOSS CURVES FOR RANDOM LR SCHEDULES WITHIN STABLE REGION

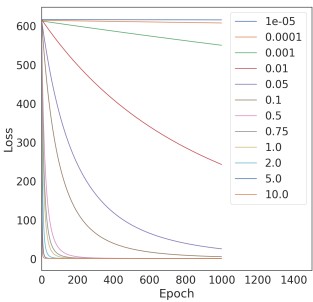

Figure 2: Loss histories corresponding to fixed LRs for NMF with SGD.

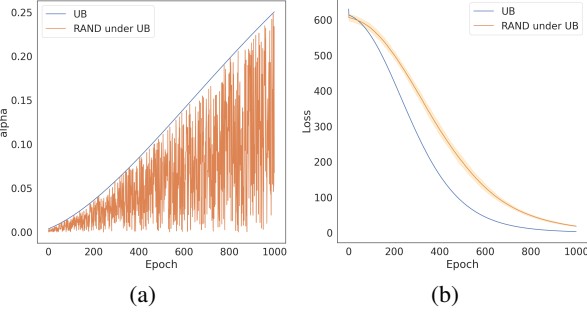

Figure 3: a) Example random walk LR schedule under the upper bound LR, b) Mean loss curve with variance corresponding to 25 experiment trials, and compared to the the upper bound loss curve.

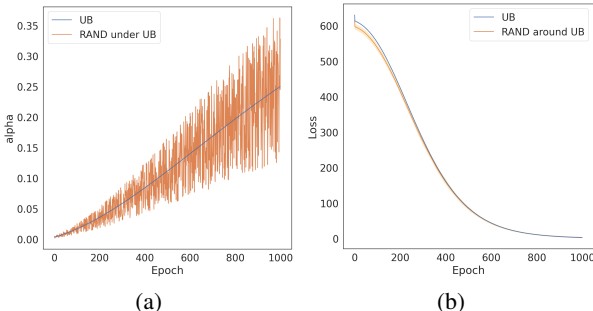

(a)                                        (b)

Figure 4: a) Example random walk LR schedule around the upper bound LR, where magnitude is $+/-50\%$ of the value on the upper bound curve, b) Mean loss curve with variance corresponding to 25 experiment trials, and compared to the the upper bound loss curve.

## C   APPENDIX C - RL FOR EFFICIENT LR SCHEDULES: TRAINING DETAILS AND RESULTS

For the NMF problem, we set the reward as the sum of

1. Percentage decrease over the best loss so far,

2. Epochs remaining, assuming a fixed computational budget for training captured a number of allowed epochs.

The agent is therefore incentivized to both reduce over the best loss so far as well as converge faster. We see that the policy converges to several local optima based on the number of RL iterations. We set the episode to end when a divergence is detected, or if loss is ¡10; this is a low enough value after which no LR changes affect the low converge rate close to an optima. This is done since we observe that even arbitrarily high LRs may slowly converge over 1000 epochs beyond a point. Below, we include loss curves and action histories of several RL experiments based on different number of RL learning iterations and RL agent Learning rates. Note that in all cases below, the PPO agent has its own LR that is fixed to some value; the action history is the LR schedule that is applied to the underlying learning problem, here - NMF.

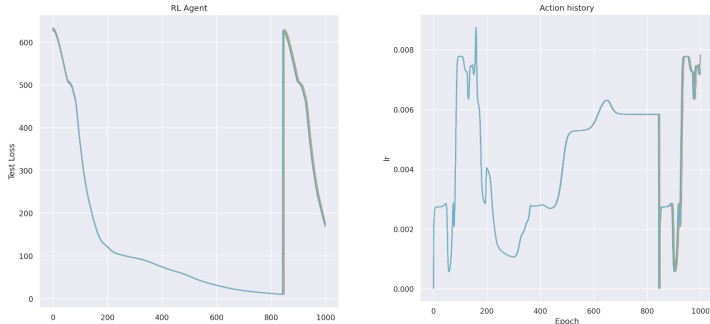

Figure 5: RL agent loss curve and action history for a PPO agent LR of 0.01 and 10K iterations. We see an interesting action history that is not a clean parametric curve, and even though the magnitude of the action is low overall, convergence rate is higher than much higher fixed LR schedules. The model converges once, resets the environment (reinitializes the problem) and starts again at around 850 epochs.

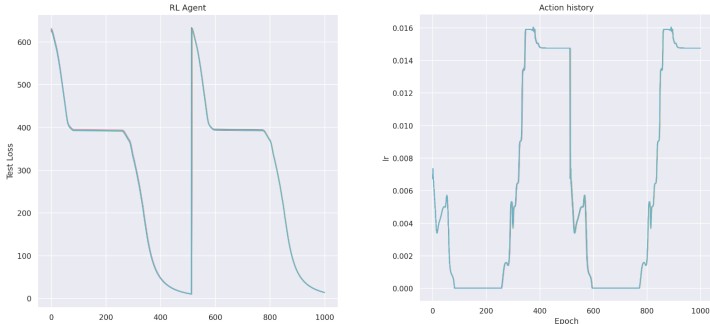

Figure 6: RL agent loss curve and action history for a PPO agent LR of 0.01 and 20K iterations. The agent converges twice and repeats the same learned schedule.

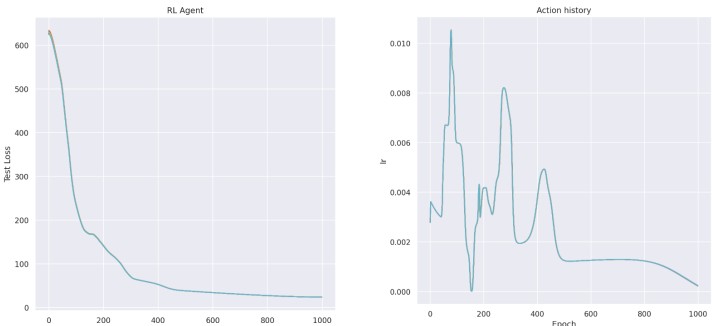

Figure 7: RL agent loss curve and action history for a PPO agent LR of 0.01 and 50K iterations. Convergence rate is lower, but looks like standard loss curves.

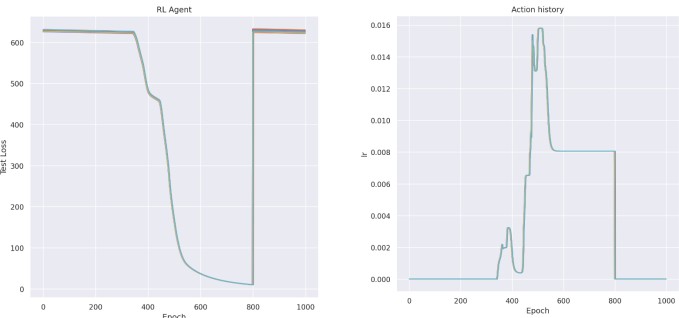

Figure 8: RL agent loss curve and action history for a PPO agent LR of 0.01 and 100K iterations. It would be interesting to understand why the agent converged to a policy where the first 500 epochs includes a fixed, lower bound LR.

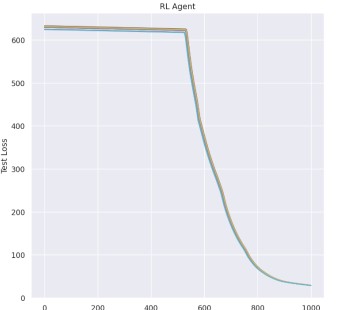 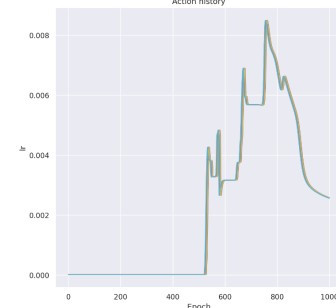

Figure 9: RL agent loss curve and action history for a PPO agent with higher LR of 0.1 and 100K iterations. It is not obvious that Higher RL learning rate and larger number of iterations may will always lead to better schedules, which would be interesting to investigate in future work.

What we conclude from these experiments is that:

1. A learned schedule may work better than a set of predetermined schedules even though the mean value of the LRs applied is lower. This is interesting since most theoretical justifications for established algorithms suggest the highest convergence rate at the highest applied LRs.

2. The agent may find several local optima based on the reward function used. So theoretically-justified reward shaping has potential to find better, more effective schedules, which we intend to investigate in future work.

3. From Figure 8 and Figure 9, we suspect that more exhaustively trained RL agents (with larger number of iterations and higher RL learning rates) could lead to sophisticated schedules where an initial phase with low learning rates is performed to adjust the policy's settings based on local gradient behavior, so that it learns how to infer second-order derivative information from the first-order observations. Then, rapid progress is attained with a series of high learning rates.

