# OpenReview forum: "Efficient Learning rate schedules for Stochastic Non-negative Matrix Factorization via Reinforcement Learning"
_ICLR.cc/2023/TinyPapers — Submitted to Tiny Papers @ ICLR 2023_

### Official Review · Reviewer_hGdf · 2023-03-24

**Confidence:** 4

**Summary Of Contributions:**

The work studies finding the optimal learning rate when alternating gradient descent is used to solve the non-negative matrix factorization problem. Mathematically the authors show a range of the learning rate lead to the algorithm convergence. The authors also propose a RL based solution to tune the learning rate and conduct experiments to visualize its performance.

**Rating:**

Great Start (GS): a submission which meets some of the reviewing criteria but has room for improvement

**Strengths And Weaknesses:**

Using learning-based methods to select a good learning rate sequence is an interesting and well-motivated problem. However, the work is still in a very early stage. The theoretical result presented in Theorem A.3 follows from simple Lipschitz continuity and PL condition and is quite elementary. The experimental setup and results are not well explained. I believe major work needs to be performed before this paper can become a good contribution to the community.

**Suggested Changes:**

1. What exactly is the loss function $F$? Is it $F(W,H)=||V-WH||_F^2$? If so, we should be able to compute the exact gradient of $F$ with respect to the two factors, which means that we can perform a deterministic gradient descent algorithm. The paper claims that SGD is performed, but I do not see how the stochasticity or noise arises in the gradient.

2. How the PL condition is established needs to be clearly explained. $F^*$ is probably the optimal function value of $\min_{W,H}F(W,H)$ but is not defined.

3. In RL application paper, it is critical to clearly define the state space, action space, and reward function. Is the action space continuous in this problem or is it discretized? The description of the reward function sounds like rewards are only given at the end of the episode, which may be very sparse. To improve RL training, I suggest designing a denser reward signal.

4. The takeaway from Figure 1 is not clear. Why is the performance of the RL policy fluctuating?

5. The operator P is undefined in the second paragraph of the introduction, which I believe is the projection to $\mathbb{R}_{+}^{m\times r}$.

---

> ### Author Response · Authors · 2023-05-31
> **Response to Reviewer hGdf (2/2)**
>
>
> ### Responses to suggested changes
>
> 1. _What exactly is the loss function F? Is it F(W,H)=||V−WH||_F^2? If so, we should be able to compute the exact gradient of F with respect to the two factors, which means that we can perform a deterministic gradient descent algorithm. The paper claims that SGD is performed, but I do not see how the stochasticity or noise arises in the gradient._
>
> **Response** : We clarified in the revised draft that we are considering a stochastic version of NMF with SGD where only unbiased noisy gradient information are available in each iteration.
>
> ------
>
> 2. _How the PL condition is established needs to be clearly explained. F* is probably the optimal function value of min_{W,H} F(W,H) but is not defined._
>
> **Response** : We incorporated the suggested change and defined F* in the revised draft.
>
> -------
>
> 3. _In RL application paper, it is critical to clearly define the state space, action space, and reward function. Is the action space continuous in this problem or is it discretized? The description of the reward function sounds like rewards are only given at the end of the episode, which may be very sparse. To improve RL training, I suggest designing a denser reward signal._
>
> **Response** : We clarified the RL training details in Appendix C of the revised draft and summarized them in the main text. We currently do not discretize the action space, and learning rate can be any float value from 1e-6 to 1. The reward function is accessed by the agent at every step of the episode. We agree that investigating a denser reward RL signal would be very interesting. Similarly, we investigated in Appendix C in the revised draft the impact of increasing the number of RL training iterations and provided a follow-up discussion. In general, whether it is information at finer resolution or more training rounds, we believe that studying the evolution of RL policy adjustments and the structure of learned policies can provide important insights.
>
> -------
>
> 4. _The takeaway from Figure 1 is not clear. Why is the performance of the RL policy fluctuating?_
>
> Response: We clarified the takeaway from the figure in the revised draft. Basically, it shows that the agent finds a schedule that leads to faster convergence than traditional algorithms that perform similarly to operating at the maximum stable learning rate (the upper bound). For the fluctuation behavior, we clarified in the revised draft - the agent resets (restarts the episode) to show that it can converge multiple times. This can be alleviated easily in practice by not allowing the agent to reset.
>
> -----------
>
> 5. _The operator P is undefined in the second paragraph of the introduction, which I believe is the projection to R_+^{m \times r}_
>
> **Response** : We removed the unnecessary detail (operator P) and just used a ‘proportional’ operator instead between the two sides.

---

> ### Author Response · Authors · 2023-05-31
> **Response to Reviewer hGdf (1/2)**
>
> Please see responses below inline:
>
> ### Responses to strengths and weaknesses
>
> **Strengths And Weaknesses** : _Using learning-based methods to select a good learning rate sequence is an interesting and well-motivated problem. However, the work is still in a very early stage. The theoretical result presented in Theorem A.3 follows from simple Lipschitz continuity and PL condition and is quite elementary. The experimental setup and results are not well explained. I believe major work needs to be performed before this paper can become a good contribution to the community._
>
> **Response** : Thank you for the meticulous review and drawing our attention to several important missing details, which helped us significantly improve the storyline and overall quality of this work. The storyline is now changed as follows: We observe that for the considered stochastic setting of NMF with SGD, it is easy to find a stable (convergence) region. However, within the stable region, traditional algorithms do not perform well at delivering high rate of convergence, mostly because (we hypothesize) they fail at effectively inferring the second-order derivative information from first-order observations. Hence, this is a clean example (as we don’t have to worry about stability regions) where we can demonstrate that it is likely that RL agents are effective at finding a policy that efficiently infers higher-order derivative information from first-order observations for the considered problem. This makes this work potentially a precursor to future works where there is a demonstrated benefit for training deep learning models, which is currently heavy on using SGD and in general, first-order derivative information. Hence, we start by the theoretical proof that stability for this problem is easy to attain, and then demonstrate the empirical evidence that RL agents deliver superior schedules.
>
> In light of this improved storyline, we would like to highlight that Theorem A.3 is merely used to show that the considered setting is a simple one where finding a stable region is not a challenge, and hence provides a clean setup for testing the effectiveness of RL in optimizing the convergence rate.
>
> ---------

---

### Official Review · Reviewer_iw8H · 2023-03-31

**Confidence:** 3

**Summary Of Contributions:**

The problem of learning problem-specific learning schedules for non-negative matrix factorization is studied. Some theoretical upper bounds for the optimal learning rate schedule are studied, and a reinforcement learning based approach is implemented and empirically studied.

**Rating:**

Great Start (GS): a submission which meets some of the reviewing criteria but has room for improvement

**Strengths And Weaknesses:**

Strengths:
- Important problem with lots of applications
- Great approach to try to obtain theoretical bounds, coupled with experiments

Weaknesses:
- Theoretical bounds need more precision
- Nice empirical demonstration, but more details about the experiment could help reproducibility

**Suggested Changes:**

Lemmas/Theorems should be stated precisely:
- Lemma A.1: statement should include convex w.r.t. which argument(s) since F is a function with two arguments.
- Lemma A.2: again argument, and also Lipschitz constant should come in lemma statement. But why is L even constant (an assumption?).

Section 2: Assumptions for any theorems should be clearly stated in the main body.

Experiments:
- How are the sparse matrices initialized?
- Details of training the RL agent are missing.
- Nice to haves: link to code; more discussion on curves in Appendix B.

Typo:
section ?? in last line Pg 1.

---

> ### Author Response · Authors · 2023-05-31
> **Response to Reviewer iw8H**
>
> Thank you for highlighting the drawbacks in the paper. We made the theoretical statements in Appendix A and their summary in the main text more precise. We have also added more details about experimental setup, and particularly the RL agent training details both in main text (as space allowed) and Appendix C. Please find the individual responses to suggested changes below.
>
>
> ### Suggested Changes (please see responses inline):
>
> _Lemmas/Theorems should be stated precisely_:
> - _Lemma A.1: statement should include convex w.r.t. which argument(s) since F is a function with two arguments._
> - _Lemma A.2: again argument, and also Lipschitz constant should come in lemma statement. But why is L even constant (an assumption?)._
>
> _Section 2: Assumptions for any theorems should be clearly stated in the main body._
>
> **Response**  : Thank you, we have incorporated the suggested changes and now clearly stated the theorems in the main text
>
> --------
>
> _Experiments_:
> - _How are the sparse matrices initialized?_
> - _Details of training the RL agent are missing._
>
> **Response** : We randomly initialize sparse matrices U and V with a density of 0.01. We have also added the details in Appendix C, and summarized them in main text.
>
> ---------
>
> - _Nice to haves: link to code; more discussion on curves in Appendix B._
>
> **Response** : We provided more insightful discussions about these curves in the main text and tried to make the captions as clear as possible in the Appendix. We will aim at sharing code in future work, but this time it was not straightforward to determine whether we can share it.
>
> ---------
>
> _Typo: section ?? in last line Pg 1._
>
> **Response** : We removed this sentence.

---

### Author Response · Authors · 2023-05-31
**Archival opt in**

Please note that we wish to opt-in for archival

---

### Meta-Review · Area_Chair_AP9u · 2023-04-02

**Recommendation:** Invite to revise
**Confidence:** 4

**Metareview:**

Strengths:
- interesting and well-motivated problem
- great principled attempt at the problem from both theoretical and empirical angles

Drawbacks:
- experimental setup and results are not well explained
- some theoretical results need more clarification (see reviewer comments)


**Summary:**

The authors study the problem of finding the optimal learning rate for alternating gradient descent for the non-negative matrix factorization problem. A range of the learning rate lead to the algorithm convergence and a RL based solution to tune the learning rate is proposed and empirically studied. Great start and direction, but theoretical and empirical results need more clarification.

**Comments And Feedback To The Authors:**

Please incorporate the feedback from the reviewers.

**Reason For Not Giving A Higher Recommendation:**

There seems to be concerns about reproducibility of experiments, and clarity of theoretical results.

**Reason For Not Giving A Lower Recommendation:**

N/A

---

> ### Author Response · Authors · 2023-05-31
> **Response to Area Chair AP9u**
>
> We appreciate the effort put into this review. The raised comments drew our attention to better understand this work and reach the right storyline for the write-up. We hope that the applied revisions and responses below will clarify the objective of this work, as we believe that it could have a positive impact on the development of future deep learning optimization algorithms. Original comments:
>
> ### Metareview:
> #### Strengths:
>
> * interesting and well-motivated problem
> * great principled attempt at the problem from both theoretical and empirical angles
>
> #### Drawbacks:
>
> * experimental setup and results are not well explained
> * some theoretical results need more clarification (see reviewer comments)
>
> Reason For Not Giving A Higher Recommendation:
> There seems to be concerns about reproducibility of experiments, and clarity of theoretical results.
>
> -------
>
> ### Response
>
> We clarified the theoretical results and particularly made the sentences more rigorous according to the reviewers’ comments, so thank you for drawing our attention. Also, note that the storyline is now changed as follows: We observe that for the considered stochastic setting of NMF with SGD, it is easy to find a stable (convergence) region. However, within the stable region, traditional algorithms do not perform well at delivering high rate of convergence, mostly because (we hypothesize) they fail at effectively inferring the second-order derivative information from first-order observations. Hence, this is a clean example (as we don’t have to worry about stability regions) where we can demonstrate that it is likely that RL agents are effective at finding a policy that efficiently infers higher-order derivative information from first-order observations for the considered problem. This makes this work potentially a precursor to future works where there is a demonstrated benefit for training deep learning models, which is currently heavy on using SGD and in general, first-order derivative information. Hence, we start by the theoretical proof that stability for this problem is easy to attain, and then demonstrate the empirical evidence that RL agents deliver superior schedules.

---

### Decision · Program_Chairs · 2023-04-08

Revision accepted; invite to archive